# Multi-state data storage in a two-dimensional stripy antiferromagnet implemented by magnetoelectric effect

Pingfan Gu[1,2], Cong Wang [3,4], Dan Su[5], Zehao Dong[1], Qiuyuan Wang[1], Zheng Han[6,7,8], Kenji Watanabe [9], Takashi Taniguchi [10], Wei Ji [3,4] ✉, Young Sun[11] ✉ & Yu Ye [1,2,8,12] ✉

A promising approach to the next generation of low-power, functional, and energy-efficient electronics relies on novel materials with coupled magnetic and electric degrees of freedom. In particular, stripy antiferromagnets often exhibit broken crystal and magnetic symmetries, which may bring about the magnetoelectric (ME) effect and enable the manipulation of intriguing properties and functionalities by electrical means. The demand for expanding the boundaries of data storage and processing technologies has led to the development of spintronics toward two-dimensional (2D) platforms. This work reports the ME effect in the 2D stripy antiferromagnetic insulator CrOCl down to a single layer. By measuring the tunneling resistance of CrOCl on the parameter space of temperature, magnetic field, and applied voltage, we verified the ME coupling down to the 2D limit and probed its mechanism. Utilizing the multi-stable states and ME coupling at magnetic phase transitions, we realize multi-state data storage in the tunneling devices. Our work not only advances the fundamental understanding of spin-charge coupling, but also demonstrates the great potential of 2D antiferromagnetic materials to deliver devices and circuits beyond the traditional binary operations.

The field of spintronics concerns the processing of digital information, where an external stimulus, preferably an electrical stimulus, is applied to control the spin order in a magnetic system that acts as a "0" or "1" digital bit. As a more common magnetic ground state than ferromagnetism, antiferromagnetism has received increasing attention in recent years due to its promising prospect for spintronic devices, such as their robustness against external perturbations, no stray fields, and

ultrafast dynamics[1]. More intriguingly, antiferromagnets often manifest complicated spin configurations and phase transitions, opening up the possibility to implement new data storage logic superior to conventional binary algorithms. However, the negligible net magnetization also makes it difficult to read out or electrically manipulate the antiferromagnetic order, which is desired for information technology. As a result, formidable efforts have been devoted to switching

[1]State Key Laboratory for Mesoscopic Physics and Frontiers Science Center for Nano-optoelectronics, School of Physics, Peking University, Beijing, China. [2]Collaborative Innovation Center of Quantum Matter, Beijing, China. [3]Beijing Key Laboratory of Optoelectronic Functional Materials and Micro-Nano Devices, Department of Physics, Renmin University of China, Beijing, China. [4]Key Laboratory of Quantum State Construction and Manipulation (Ministry of Education), Renmin University of China, Beijing, China. [5]Beijing National Laboratory for Condensed Matter Physics, Institute of Physics, Beijing, China. [6]State Key Laboratory of Quantum Optics and Quantum Optics Devices, Institute of Optoelectronics, Taiyuan, China. [7]Collaborative Innovation Center of Extreme Optics, Shanxi University, Taiyuan, China. [8]Liaoning Academy of Materials, Shenyang, China. [9]Research Center for Functional Materials, National Institute for Materials Science, Tsukuba, Japan. [10]International Center for Materials Nanoarchitectonics, National Institute for Materials Science, Tsukuba, Japan. [11]Center of Quantum Materials and Devices, and Department of Applied Physics, Chongqing University, Chongqing, China. [12]Yangtze Delta Institute of Optoelectronics, Peking University, Nantong, China. ✉e-mail: wji@ruc.edu.cn; youngsun@cqu.edu.cn; ye_yu@pku.edu.cn

antiferromagnets through spin torques from exchange bias[2], spin-orbit torque[3] and spin-galvanic effect[4], etc.

Apart from the above-mentioned "external" approaches, the "internal" approach to antiferromagnetic spintronics rests upon materials with coupled degrees of freedom[5–7] such as magnetic moment, electric polarization and strain, which are often referred to as the magnetoelastic or magnetoelectric materials[8]. However, as the transition metal $d$ electrons are supposed to repel the tendency for off-center ferroelectric distortion[9], the coexistence of magnetic moment and electric polarization is hard to achieve. The coupled energy terms, as well as the electric polarization, require materials with low crystal and magnetic symmetry. Improper magnetic ferroelectrics[10,11] where the ferroelectricity originated from spin-order-driven inversion symmetry breaking are considered to be an ideal platform to realize the mutual clamping of the order parameters of ferroelectricity and antiferromagnetism[12–14]. Based on the spin frustration and spin-orbit coupling theories, three spin-structure-induced electric polarization mechanisms have been established, namely the exchange striction model[15], the inverse Dzyaloshinskii-Moriya interaction model[16] and the $p$-$d$ hybridization model[17].

To further explore ME-based fundamental physics and develop practical applications for information devices, it is inevitable to extend the research to the two-dimensional (2D) limit[18–23]. Recently, multiferroicity resulting from inverse Dzyaloshinskii-Moriya interaction[24,25] and $p$-$d$ hybridization[26] have been evidenced in van der Waals materials. Nevertheless, the direct coupling between magnetic moment and electric polarization in the 2D limit has not been reported. The air-stable van der Waals insulator CrOCl, with a stripy antiferromagnetic ground state and thus both broken rotational symmetry and translation symmetry perpendicular to the stripes[27], appears to be a potential candidate to exhibit the intrinsic 2D ME effect. The magnetic phase transitions and magnetoelastic coupling effect of CrOCl have been verified in the 2D limit[27,28]

previously. In addition, exotic quantum Hall effects[29] and insulator phases[30] were observed in the graphene/CrOCl heterostructures, implying rich physics in this fascinating material.

In this work, we report the intrinsic ME effect in CrOCl down to the 2D limit by tunneling transport measurement. The $I-V$ curve of the CrOCl tunneling device exhibits obvious hysteresis below the Néel temperature, signaling an electric phase transition accompanying the magnetic phase transition. The metastable states are supposed to result from the antiferroelectric dipoles in the low-temperature phase, which was evidenced by first-principles calculations and dielectric measurements. We also realize direct manipulation of resistance states through electric and magnetic field sweeps, demonstrating a data storage device with continuously adjustable outputs. The metastable states in the first-order phase transition resulting from the ME coupling break through the barriers of write-in and read-out information in antiferromagnets and hopefully can open an effective route for multi-state memories with high storage density, nonvolatility, low energy consumption, and small memory cell.

## Results and discussion

To understand the intrinsic ME effect, we start with the energy coupling terms for the ground state of CrOCl. As reported by several neutron scattering results[31–33] and confirmed by DFT calculations[27,34], CrOCl transforms into the ↑↑↓↓ stripy AFM ground state at ~ 14 K. The magnetic frustration drives the crystal from the orthorhombic space group *Pmmn* (Fig. 1a) to the monoclinic space group *P2₁/m* (Fig. 1b)[32,33,35]. Despite the monoclinic distortion, the space inversion operation *P* preserves in the ↑↑↓↓ ground state due to the special zigzag atom-chain structure of CrOCl. The two sublayers of a single van der Waals layer are spatially inverse to each other, thus preventing spontaneous electric polarization from appearing. However, based on the symmetry operations of stripy antiferromagnets, we can obtain the

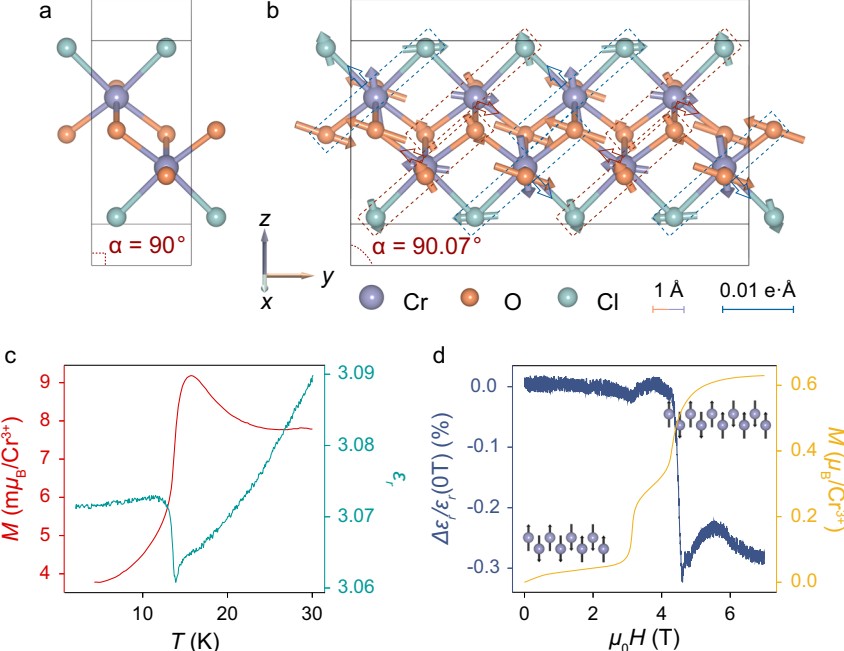

**Fig. 1 | Magnetic and dielectric properties of bulk CrOCl. a** Crystal structure of CrOCl above the Néel temperature with the *Pmmn* space group in a unit cell. **b** Crystal structure of CrOCl below the Néel temperature with the *P2₁/m* space group in a unit cell. The structural parameter $\alpha$ that defines the crystal symmetry is labeled in the unit cell. The vectors on each atom represent the atomic

displacements (magnified 100 times), while the dashed boxes and open arrows represent the calculated antiferroelectric polarizations of different Cl-Cr-O chains. **c** Temperature dependence of relative permittivity $\varepsilon_r$ and magnetic moment of bulk CrOCl crystal. **d** $\Delta\varepsilon_r/\varepsilon_r(0\ \text{T})$ *versus* out-of-plane magnetic field and the corresponding $M-H$ curve at 2 K.

free energy in the wave form[36,37]:

$$\mathcal{F} = \frac{1}{2} r_a |A_{\mathbf{k}}|^2 + \frac{1}{2} r_s |\mathbf{S}_{\mathbf{q}}|^2 \\ + \lambda_1 \left[ (\mathbf{S}_{\mathbf{q}} \cdot \mathbf{S}_{\mathbf{q}}) A_{\mathbf{k}}^* + \text{c.c.} \right] + 2\lambda_2 |\mathbf{S}_{\mathbf{q}}|^2 |A_{\mathbf{k}}|^2 \qquad (1)$$

where $\mathbf{S}_{\mathbf{q}}$ is the amplitude of spin density modulation with the wave vector of $\mathbf{q}$ and $A_{\mathbf{k}}$ is the corresponding lattice modulation with the wave vector of $\mathbf{k}$. $r_a$, $r_s$, $\lambda_1$ and $\lambda_2$ are constant parameters of the material. Minimizing the free energy, we obtain the lattice modulation in a collinear spin configuration:

$$A_{\mathbf{k}} = \frac{\lambda_1}{r_a} \mathbf{S}_{\mathbf{q}} \cdot \mathbf{S}_{\mathbf{q}} + \text{c.c.} \qquad (2)$$

The lattice modulation, consequently, can only be present with $\mathbf{k} = \pm 2\mathbf{q}$. In other words, a long-range magnetic wave order with a period of $4b$ ($b$ is the lattice constant along the crystal $b$-axis) induces a strain wave with half of its period, $2b$. The displacements of the atoms in CrOCl have been verified and resolved by neutron scattering[32,33] and X-ray diffraction experiments[35]. Our density functional theory (DFT) calculations also confirm the displacements of atoms in the ↑↑↓↓ ground state with a period of $2b$, (Fig. 1b), showing similar displacement directions and magnitudes as the neutron scattering experiments[33]. Based on the above phenomena, we can regard the ↑↑↓↓ stripy AFM phase as an "antiferroelectric" order consisting of the electric dipoles of the distorted Cl-Cr-O chains away from the inversion center (dashed boxes in Fig. 1b). Therefore, ordered effective dipoles are formed as indicated by the open arrows in Fig. 1b, while the electric dipole of each atom is provided in Supplementary Information Fig. S1. Both the contributions of ions and electrons to polarization are taken into account by the Born effective charge method (see Methods).

To further investigate the two-fold distorted order and the ME effect, we conducted dielectric and pyroelectric measurements in CrOCl crystals. The pyroelectric current exhibits a small but observable peak at the Néel temperature, which can be attributed to the antiferroelectric domains locally breaking the inversion symmetry (see detailed discussions in Supplementary Information Fig. S2 and S3). Correspondingly, the relative permittivity $\varepsilon_r$ of CrOCl shows a sharp increase of ~0.01 at ~14 K (the $\varepsilon_r - T$ curve in Fig. 1c) with decreasing temperature. It should be noted that the layer thickness experiences a sudden expansion at the Néel temperature[32], which tends to result in a decrease in the measured capacitance, as $C = \frac{\varepsilon_r \varepsilon_0 S}{d}$ (see Methods). As a result, the increase in $\varepsilon_r$ can only be attributed to the appearance of additional tunable dipoles. Most importantly, both the pyroelectric peaks and the change of $\varepsilon_r$ completely accord with the Néel temperature where the net moment drops (the $M - T$ curve in Fig. 1c), evidencing that the structural phase transition is a product of ME coupling. In other words, the symmetry-allowed ME coupling term guarantees that there is a corresponding relationship between the magnetic order and the electric order, so when the magnetic order experiences phase transitions, the electric order transforms correspondingly. This is the basic concept of the ME effect in CrOCl, which is also manifested in the phase transitions under the magnetic field (Fig. 1d).

According to the previous report[27], CrOCl transforms into the collinear ↑↑↑↓↓ phase under an external out-of-plane field of ~4.5 T (the $M - H$ curve in Fig. 1d). Our calculations show that in the ↑↑↑↓↓ phase, the crystal relaxes to an orthorhombic structure (see Supplementary Information Table S1), in which the atomic displacements and the electric dipoles show a period of $5b$. Under an external electric field of 0.07 V/Å (close to the experimental values of the tunneling measurements below), the ↑↑↓↓ phase produces a net polarization of 116.50 μC/m², which is eight times larger than that of the ↑↑↑↓↓ phase (14.42 μC/m²). Correspondingly, the measured $\varepsilon_r$ undergoes a decrease upon the magnetic phase transition to the ↑↑↑↓↓ phase (Fig. 1d), so we can simply estimate the additional polarization from the dielectric measurement, $\Delta P = \Delta \varepsilon_r \cdot \varepsilon_0 E \approx 61.98$ μC/m², very close to the polarization difference between the ↑↑↓↓ and ↑↑↑↓↓ phases of ~102.11 μC/m² under the same electric field. This suggests that in the ↑↑↓↓ structurally distorted phase, there is another mechanism that produces the additional electric polarization under an external field, possibly the canting of the antiferroelectric dipoles. On the contrary, the ↑↑↑↓↓ phase is rather structurally stable and the electric dipoles can hardly be adjusted by the external field.

Now that we have demonstrated the existence of spin-induced additional electric polarization in bulk CrOCl, from a macroscopic perspective, we would expect the interactions between the electric and magnetic degrees of freedom and consequently the manipulation of the spin order by the electric field, or vice versa. To achieve a large electric field in ultra-compact nanodevices, we fabricated vertical tunneling devices based on single- to few-layer CrOCl flakes (Fig. 2a). Cross-structured few-layer graphite stripes are used to contact the CrOCl tunneling layer, and the whole device is encapsulated by hexagonal boron nitride ($h$-BN). The electric field in CrOCl can reach ~0.1 V/Å, where the spin-induced electric dipoles are expected to be polarized and have a substantial impact on the tunneling current.

We first demonstrate the influence of the additional electric polarization in CrOCl flakes through the hysteresis of the $I-V$ curves. Here we show the tunneling current of a CrOCl device with a thickness of ~9.1 nm (approximately 12 layers), labelled as device 1. As shown in Fig. 2b, the tunneling currents in the sweep-up and the sweep-down processes deviate significantly from each other under the same applied voltage at a temperature of 2 K, manifesting as an obvious clockwise hysteresis. In sharp contrast, the $I-V$ curve at 20 K (above the Néel temperature) shows no sign of hysteresis. The electric hysteresis can be further viewed in the 2D map of temperature and bias voltage (Fig. 2c), where the current polarization $\rho$ is defined as $\rho = (R_{\text{up}} - R_{\text{down}})/(R_{\text{up}} + R_{\text{down}})$. Clearly, $\rho$ exhibits non-zero values below ~12 K, in perfect accordance with the Néel temperature of the exfoliated CrOCl[27]. The clockwise hysteresis results in a negative $\rho$ under positive bias and a positive $\rho$ under negative bias. The absolute value of $\rho$ decreases with increasing temperature and eventually becomes zero at the Néel temperature (Fig. 2c, d), implying the disappearance of the additional polarization as the antiferromagnetic order collapses. From the $R - T$ curves of different sweeping processes at 5 V (Fig. 2e), it can be seen that in the sweep-up process, the resistance drops significantly (entering a lower resistance state) below the Néel temperature, while in the sweep-down process, the resistance increases (entering a higher resistance state). The above phenomena indicate that the direction of the additional polarization, in other words, the configuration of the dipole order, dominates in the change of the measured tunneling resistance upon the magnetic phase transition. The resistance at −5 V exhibits almost antisymmetric behavior, as shown in Supplementary Information Fig. S4.

As mentioned earlier, in the absence of adjustable electric dipoles, the ↑↑↑↓↓ phase under an external magnetic field is rather structurally stable. The resistances under different electrical sweeping processes converge at the transition field (Fig. 2f) and show completely different magnetoresistance behaviors at the ↑↑↑↓↓ phase (see details in Supplementary Information Fig. S5 and S6), which implies the disappearance of the $I-V$ hysteresis in the ↑↑↑↓↓ phase and is in perfect agreement with our theory. The hysteresis of the $I-V$ curve in the ↑↑↓↓ state can be observed in all CrOCl devices of varying thicknesses (Supplementary Information Fig. S7) down to monolayer. Suspending the sweeping process at a specific voltage, the resistance value relaxes

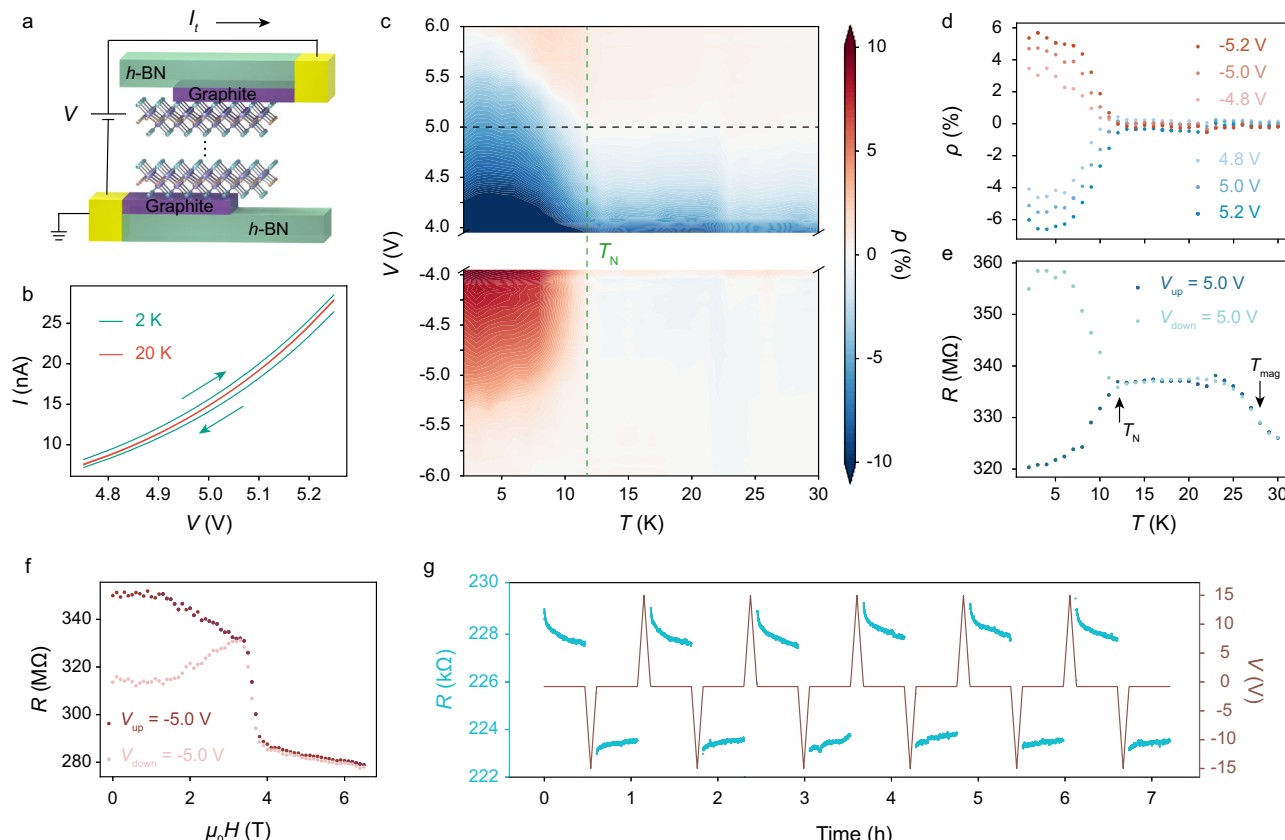

**Fig. 2 | I − V hysteresis of CrOCl tunneling devices. a** Illustration of the tunneling device. **b** I − V curves of device 1 at 20 K (above the Néel temperature) and 2 K (below the Néel temperature). **c** 2D map of current polarization ρ as a function of temperature and bias voltage. The green dashed line marks the Néel temperature of the exfoliated CrOCl. **d** ρ versus temperature at different voltages extracted from **b**. **e** Resistance versus temperature of the CrOCl tunneling device at 5 V in different sweeping processes. **f** Tunneling resistance versus out-of-plane magnetic field at − 5 V in different sweeping processes. **g** Reproducible manipulation of the resistance states at − 0.8 V by alternately changing the sweeping direction. Data in **a**–**f** were obtained in device 1 ( ~ 9.1 nm CrOCl), while **g** was obtained in device 2 (monolayer CrOCl) in parallel with a 10 MΩ protection resistor.

slowly with a time constant of several hours (see Supplementary Information Fig. S12), which is highly reproducible as shown in a single-layer CrOCl device (device 2 in Fig. 2g).

We note that in tunneling devices, extrinsic factors such as electromigration, oxygen vacancy redistribution, or charge trappings may also result in such hysteresis behaviors[38]. However, the fact that the hysteresis accompanies the magnetic transitions distinguishes it as an intrinsic characteristic of CrOCl. In addition to Fig. 2, we present the 2D map of ρ over the entire voltage range (Supplementary Information Fig. S8 and S9), the hysteresis behavior in different devices (Supplementary Information Fig. S7) and in different temperature-sweeping processes (Supplementary Information Fig. S10) to exclude the possible extrinsic origins. Consequently, the additional polarization, or the tilting of electric dipoles is supposed to be the most plausible physical mechanism to explain the I − V hysteresis. Holding the simplest idea that the external field induces an additional polarization from the antiferroelectric order and thus modifies the tunneling barrier, we employed a tunneling model to explain the hysteresis behaviors of the I − V curve (see calculation details in Supplementary Information Fig. S13). The graphite stripes we used as contact electrodes are not perfectly metallic, so the surface charges induced by the net polarization are not completely screened by the graphite stripes and the depolarization electric field is therefore not zero[39,40]. Based on the material parameters, we modeled the redistributed energy profile and simulated the I − V curve of the CrOCl tunneling device. The simulated current polarization ρ is ~ 1%, perfectly reproducing our experimental results, so we may obtain a thorough understanding of

the hysteresis and relaxation behaviors in CrOCl tunneling devices (Supplementary Information Fig. S11-13). The ramping electric field acts as an electric excitation to tilt the electric dipoles, resulting in a higher tunneling current, while the interactions between spin and charge tend to relax the system to the antiferroelectric ground state with a lower current. In addition to the I − V hysteresis, the structural modulation of the ↑↑↓↓ phase under the electric field also leads to different magnetoresistance behavior with varying bias voltage, as shown in Supplementary Information Fig. S14 and S15.

Switching the tunneling resistance between different metastable states resulting from ME coupling by sweeping the electric field provides us with design principles for magnetoelectronic devices. We have demonstrated that the magnetic transitions in CrOCl under external fields are first-order transitions with large hysteresis loops[27]. Therefore, we can expect that the electric field can adjust the tunneling resistance to any meta-stable states inside the magnetic hysteresis loop by tilting the electric dipoles. From the I − B curve of tunneling device 3 ( ~ 11.3 nm) at 5.5 V (Fig. 3a), the hysteresis closes at $B_s$=5.63 T, where CrOCl is believed to transform to the ↑↑↑↓↓ state. Subtracting the current of the B-up curve from that of the B-down curve, we obtain the maximum hysteresis occurs at $B_0$=3.33 T (Fig. 3a). In other words, the spin configuration of CrOCl at 3.33 T is close to the ↑↑↓↓ AFM ground state (low current) in the B-up cycle, but remains in the ↑↑↑↓↓ state (high current) in the B-down cycle. The maximum value of δI is ~ 2 nA, 10% of the tunneling current. Similar to the operation in Fig. 2, we apply electric excitations by sweeping the bias voltage up and back to 5.5 V. After the electric excitation, the tunneling current changes to

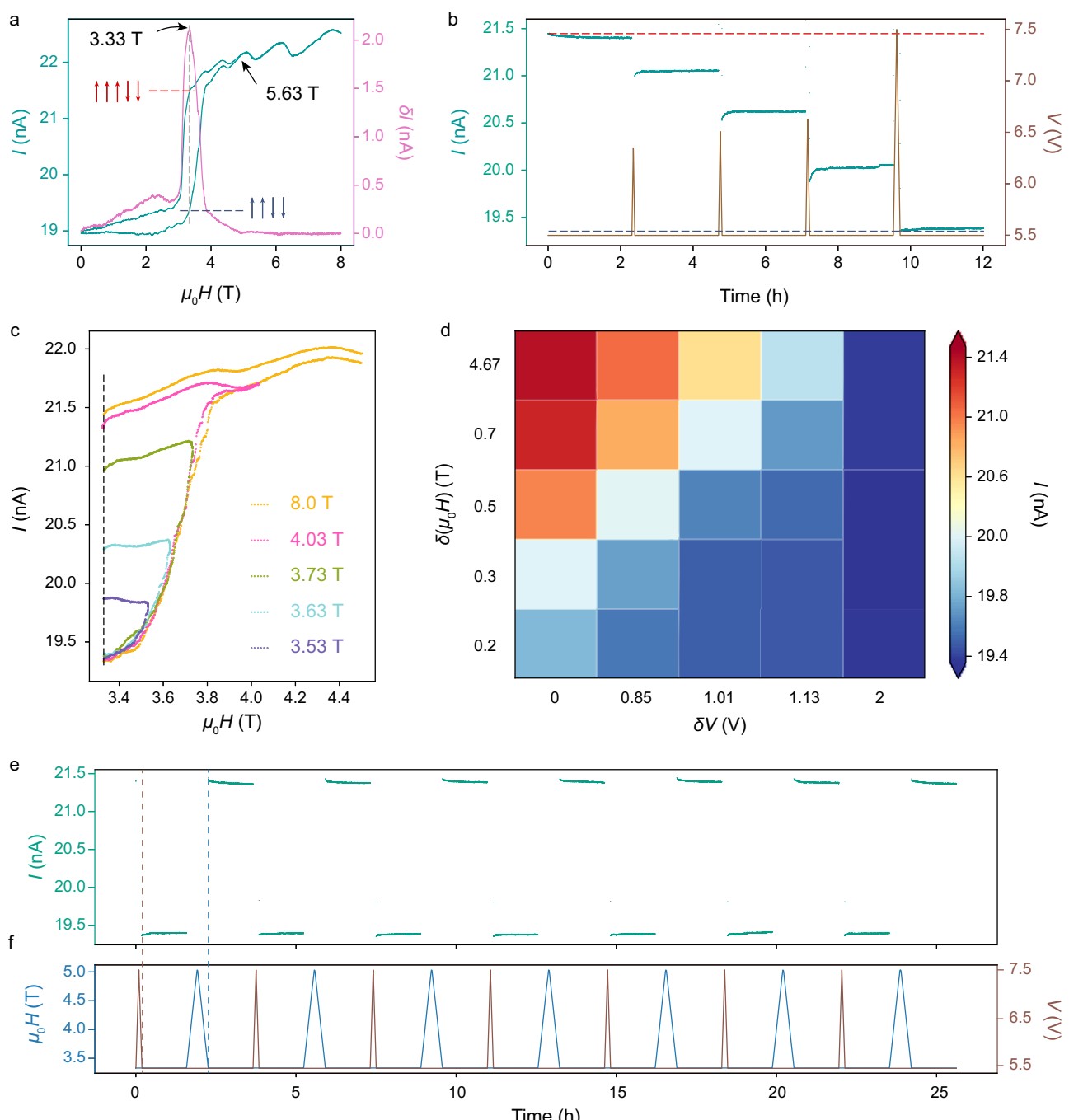

**Fig. 3 | Operation principles of multi-state data storage. a** $I − B$ curve of the CrOCl tunneling device at 2 K. The pink curve shows the differential current $\delta I = I(B)|_{up} − I(B)|_{down}$. The critical field where $\delta I$ reaches a maximum (at $B_0$) and the hysteresis closes (at $B_s$) are annotated by black arrows. **b** Tunneling current over time after several electric sweeps. The dashed lines represent the current values at 3.33 T in $B_{down}$ and $B_{up}$ sweeping cycles extracted from **a**. **c** $I − B$ curves with different magnetic sweep ranges. **d** 2D map of the tunneling current after designed electric and magnetic sweeps. **e, f** Tunneling current **e** after alternating electric and magnetic sweeps. The corresponding magnetic and electric fields *versus* time are plotted in **f**. Data were obtained in device 3 ( ~11.3 nm CrOCl).

an intermediate state following a slight relaxation and stabilizes at the state for several hours without any sign of change. The current value of the intermediate state is determined by the electric excitation, specifically, by the sweeping rate and the peak voltage. By applying different electric excitations by design, we can switch the tunneling current to an arbitrary value between the highest (↑↑↑↓↓) and lowest (↑↑↓↓) current values, as demonstrated in Fig. 3b.

It is worth noting that the relaxation process of the intermediate in Fig. 3b is much less than that at 0 T (Fig. 2g and Supplementary Information Fig. S12). This is because the external magnetic field has

overcome the energy barrier of the magnetic ground state and brings the spin configuration of CrOCl into a series of metastable states. As a result, no longer will the ME coupling energy relaxes the crystal to the antiferroelectric ground state. In other words, both the magnetic order and the electric order are amongst a series of metastable states and the system has entered neutral equilibrium, where the resistance state can be manipulated both by the electric field and the magnetic field. The noise fluctuation of each state is less than 0.02 nA, which means that a single tunneling junction can store at least a decimal number if the difference between distinguishable adjacent states is set to be an order

of magnitude larger than the noise fluctuation. After sweeping the magnetic field across $B_s$ and back to $B_0$, the junction can be reset to the $\uparrow\uparrow\downarrow\downarrow$ high current state, which means the erasing of the stored information.

Likewise, to expand the capabilities of the device, the magnetic field can serve as another degree of freedom to tune the tunneling resistance. Sweeping the magnetic field to a value below $B_s$ and back to $B_0$ also adjusts the tunneling current to an intermediate state, as shown in Fig. 3c. Accordingly, we can adjust the tunneling current to different values with varying magnetic excitations, and set the current back to the $\uparrow\uparrow\downarrow\downarrow$ low current state with a large electric excitation as well. Repeating the previous electric sweeps following different magnetic sweeps, we obtain a new set of current values. In this way, we constructed a $5 \times 5$ 2D list of different resistance states by electric and magnetic excitations (Fig. 3d). The opposite dependence of the current on electric and magnetic excitations strongly evidences the ME coupling in this system, as in the classical electrodynamics picture, the magnetic field flips the electron spins and the electric field reverses the electric dipoles. Following the counterclockwise hysteresis nature of first-order transitions, the intermediate state should be more polarized along the control parameter than the initial state. At $B = 3.33$ T, the magnetic moment of the $\uparrow\uparrow\uparrow\downarrow\downarrow$ phase is definitely larger than that of $\uparrow\uparrow\downarrow\downarrow$ phase, while the electric polarization is larger in the $\uparrow\uparrow\downarrow\downarrow$ phase at $V = 5.5$ V due to the fact that the additional polarization disappear in the $\uparrow\uparrow\uparrow\downarrow\downarrow$ phase. As a result, sweeping the magnetic field can drive CrOCl from the $\uparrow\uparrow\downarrow\downarrow$ state into the intermediate states, and symmetrically, sweeping the electric field can drive CrOCl from the $\uparrow\uparrow\uparrow\downarrow\downarrow$ state into the intermediate states.

We finally verified the repeatability of device operation as shown in Fig. 3e, f. The device is first set to the $\uparrow\uparrow\downarrow\downarrow$ AFM ground state ($\sim 19.4$ nA) by a sufficiently large electric sweep of $\delta V = 2$ V, and then reset to the $\uparrow\uparrow\uparrow\downarrow\downarrow$ state ($\sim 21.4$ nA) by the magnetic sweep of $\delta B = 1.67$ T. The writing and erasing operations were repeated seven times, and the high and low resistance states showed perfect stability. Furthermore, it is worth mentioning that similar multi-state data storage operation principles are also applicable at zero-field and to single-layer CrOCl devices (see Supplementary Information Fig. S16-18 and following discussions). Due to the fact that the single-layer CrOCl shares the same magnetic and electric order as the multi-layer CrOCl (Supplementary Information Fig. S16), the ME coupling can also persist down to the monolayer. The resistance of the single-layer tunneling device can be set to two values by applying magnetic and electric sweeps, as shown in Supplementary Information Fig. S17. The only distinction in single-layer is that there are fewer metastable states so the adjustable resistance range is lower, probably due to the thinner tunneling barrier and the absence of the interlayer dipole interactions.

We should emphasize that, although the picture of tilting antiferroelectric dipoles under an electric field seems to be the most reasonable explanation for the multi-level resistance and ME coupling, it is so far a theoretical hypothesis and needs further direct experimental evidence. Similar multi-level states as CrOCl have been reported in several other systems utilizing different mechanisms, such as the metal-insulator transition in $VO_2$[41–43], ferroelectric switching in $BaTiO_3$[44,45], and molecule trapping in $WSe_2$[46]. Although the operation of CrOCl tunneling devices is similar to that of $VO_2$ and $BaTiO_3$, the underlying mechanisms are substantially different. Only one control parameter, the electric field, can manipulate the one-dimensional phase transitions in the previously reported materials. In contrast, the ME effect in CrOCl enables the manipulation of tunneling resistance in two degrees of freedom, the electric field and magnetic field, as illustrated in Fig. 3d. We can write in the information either by electric excitations or magnetic excitations and erase the information by another. As a result, the multi-level resistance manipulation in CrOCl is not a simple extension of the new-type memristor, but a breakthrough in the fundamental concept of spintronics. It is also attractive to consider the prospects of CrOCl compared to the conventional spintronic devices. CrOCl is air-stable and can be easily exfoliated into a 2D form, providing much convenience for device fabrication. The intralayer antiferromagnetic order of CrOCl prohibits stray fields, which means that the area of the tunneling junction can be as small as possible. As the operating current is $\sim 10^1$ nA, the writing current density in our reported device is $\sim 10^0$ A/cm$^2$. The device is constructed entirely of van der Waals materials, so the device thickness can be controlled at the atomic scale. Taken together, the CrOCl tunneling device can contain ultra-high information density with ultra-low energy consumption utilizing the continuously adjustable outputs, and shows greater potential in future data-storage applications compared to previously reported materials.

In summary, we have demonstrated the ME effect in the 2D stripy antiferromagnet CrOCl. By means of dielectric measurements, tunneling measurements, and first-principles calculations, we verified the additional contribution of electric polarization below the Néel temperature, which is possibly induced by the adjustable antiferroelectric dipoles. The ME coupling term gives rise to successive metastable states that are rather stable and hardly degenerate over time, so the tunneling resistance of CrOCl can be set to arbitrary values via electric and magnetic excitations. The multi-state data storage realized in CrOCl may serve as a new paradigm with the potential to impact information technology, such as analog data storage and computation in an array of tunneling devices, stepping beyond Von Neumann architecture and enabling neuromorphic computing with low power consumption. Furthermore, the ME coupling in CrOCl may give rise to more unexplored fantastic properties, highlighting the characteristics of 2D antiferromagnetic materials and their promising potential in fundamental research and spintronic applications.

## Methods

### Crystal synthesis and characterizations
The mixture of powdered $CrCl_3$ and $Cr_2O_3$ with a molar ratio of 1:1 and a total mass of 1.5 g was sealed in an evacuated quartz ampule. The ampule was then placed in a two-zone furnace where the source and sink temperatures for growth were set to 940 °C and 800 °C, respectively, and kept for two weeks. Subsequently, the furnace was slowly cooled to room temperature, and high-quality CrOCl crystals were obtained. Magnetization measurements were performed by standard modules of a Quantum Design PPMS.

### Dielectric and pyroelectric measurements
Both pyroelectric measurements and dielectric measurements were performed on a TeslatronPT System, Oxford Instruments. Silver epoxy was painted on opposite sides of the sample as electrodes. For pyroelectric measurements, an electrometer (Keithley 6517B) was used as a DC voltage source and ammeter. When the temperature was stabilized at 60 K, an external electric field was applied to the sample. The sample was then cooled from 60 K to 2 K under different electric fields. After the temperature was stabilized at 2 K, the electric field was removed and the temperature was increased to 60 K at a rate of 5 K/min. During the heating process, the change of pyroelectric current with temperature was collected. The polarization can be obtained by integrating the pyroelectric current with time. Dielectric measurements were made using a capacitive bridge meter (AH2700A). From 2 K to 50 K, the change in relative permittivity with temperature was measured by heating at the rate of 2 K/min. Magneto-dielectric effect at 2 K was measured at a magnetic field sweep rate of 15 Oe/s from 0 to 8 T. All test frequencies are 20 kHz.

### Device fabrication
Few-layer graphite, $h$-BN (10–30 nm), and CrOCl flakes were obtained by the scotch tape exfoliation method under ambient conditions. The heterostructures were then assembled with a conventional

pick-up-and-stack technique based on polypropylene carbonate(PPC)/polydimethylsiloxane(PDMS) polymer stacks placed on glass slides. Once encapsulated, the devices were annealed in a high vacuum with a mixed gas flow of $H_2$ and Ar to remove residual PPC. Metal contact of Cr/Au (5/25 nm) electrodes were then defined using electron-beam lithography, reactive ion etching (in plasma of the $CHF_3/O_2$ mixture), electron beam evaporation, and lift-off processes.

## Electrical transport measurements

Transport measurements were performed in a Heliox$^3$He insert system equipped with a 14 T superconducting magnet. The lowest temperature of the system is 1.6 K. To measure the $I-V$ characteristics of the tunnel barrier and the magnetoresistance, a Keithley 2636B source meter was used to apply a bias voltage and a standard two-probe module was used to measure the tunneling current. To obtain intrinsic signals and at the same time exclude the possibility of the Joule heating effect, the tunneling current is limited to ~ 50 nA, so the total power in a junction of ~ $1\,\mu m^2$ is merely ~ 0.3 μW.

## DFT calculations

Our DFT calculations were performed using the generalized gradient approximation for the exchange-correlation potential, the projector augmented wave method[47], and a plane-wave basis set implemented in the Vienna ab-initio simulation package (VASP)[48]. Dispersion correction was made at the van der Waals density functional (vdW-DF) level[49], with the optB86b functional for the exchange potential[50], and which was proved to be accurate in describing the structural properties of layered materials[51] and was adopted for structure related calculations. The shape and volume of each supercell and all atomic positions were fully relaxed until the residual force per atom was less than $1 \times 10^{-3}\,eV\,Å^{-1}$ in our calculations. In VASP calculations, the kinetic energy cut-off for the plane-wave basis set was set to be 700 eV for geometric and electronic structure calculations. A $k$-mesh of $10 \times 14 \times 4$ was adopted to sample the first Brillouin zone of the conventional unit cell of CrOCl bulk. The $U$ and $J$ values of the on-site Coulomb interaction of the Cr $d$ orbitals are 3.0 eV and 1.0 eV, respectively, as revealed by a linear response method[52] and comparison with the experimental results[27]. These values are comparable to those adopted in modeling CrSCl[53] and CrI$_3$[54]. The Born effective charges were calculated using density functional perturbation theory[55]. The dipole moment of each atom is calculated by $P_i = Z_i^* \cdot \mu_i$, where $P_i$ is the dipole moment of the ion $i$ in one unit cell, $Z_i$ is the Born effective charge tensors and $\mu_i$ is the atomic displacement[56]. For calculations of single-layer CrOCl, a sufficiently large vacuum layer over 20 Å along the out-of-plane direction was adopted to eliminate the interaction among monolayer. The out-of-plane electric polarization of single-layer CrOCl under the external electric field is well defined in terms of the classical model due to the presence of a vacuum layer, which is calculated by integrating electron density times $z$-coordinate over the supercell.

## Data availability

The source data generated in this study have been deposited in the Zenodo database under the accession code https://doi.org/10.5281/zenodo.7890630. Source data are provided with this paper.

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

## Acknowledgements

This work was supported by the National Key R&D Program of China (Grants Nos. 2022YFA1203904, 2022YFA1203902, 2021YFA1400300 and 2018YFA0306900), the National Natural Science Foundation of China (No. 12250007), and Beijing Natural Science Foundation (Grant No. JQ21018). Y.S. acknowledges support from the National Natural Science Foundation of China (Grant No. 51725104). W.J. acknowledges support from Strategic Priority Research Program of the Chinese Academy of Sciences (Grant No. XDB30000000), and the National Natural Science Foundation of China (Grants No. 11974422 and No. 12104504). C.W. was supported by the China Postdoctoral Science Foundation (2021M693479). Calculations were performed at the Physics Lab of High-Performance Computing of Renmin University of China, Shanghai Supercomputer Center. T.T. acknowledges support from the JSPS KAKENHI (Grant Nos. 19H05790 and 20H00354) and A3 Foresight by JSPS.

## Author contributions

Y.Y. and P.G. conceived the project. P.G. synthesized the CrOCl crystals and fabricated the devices. P.G. conducted the transport measurements with the help of Z.D., Q.W., and Z.H. C.W. conducted the DFT calculations under the supervision of W.J. D.S. conducted the dielectric and pyroelectric measurement under the supervision of Y.S. K.W. and T.T. grew the *h*-BN bulk crystals. P.G. and Y.Y. drafted the manuscript. All authors discussed the results and contributed to the manuscript.

## Competing interests

All authors declare the following competing interests: Chinese patent (no. 202210324321.1) for using CrOCl to implement multi-state data storage in tunneling devices, which is now under consideration.
