## [Peer Review File · Nature Communications]

Reviewers' Comments:

Reviewer #1:

Remarks to the Author:

First of all, I would like to thank the authors for their careful and extensive answers to my questions (also questions from the other two reviewers). Overall, the manuscript presents an interesting magnetoelectric (ME) coupling in 2D CrOCl, where electric field can be used to control the magnetic order. Therefore, I support the publication of the paper.

Regarding some details of the reply, however, I would like to give some replies to the authors and perhaps also give some suggestions about how to improve the manuscript. I don't think the authors have presented direct evidence for the antiferroelectric (AFE) order. Only the two fold lattice distortion has been previously observed by neutron and XRD. But the two fold lattice distortion is only a prerequisite for the AFE order. The AFE order itself, remains to be directly confirmed. On the other hand, I don't think the AFE order is crucial for the key experimental observations, which is the ME coupling, i.e., electric field can directly control the magnetism in 2D CrOCl. I understand that, in the current understanding, the authors need the AFE order to explain the mechanism for the ME coupling: the authors believe that E-field couples to the AFE order, which in turn couples to the magnetism. However, to me, that is only an interpretation not a direct observation. Therefore, I suggest them to tone down their claim of the AFE order. They can focus on presenting their observations of how E and B fields control the magnetic order. Then they can suggest that the observed ME coupling is due to the existence of an AFE order. Overall, the authors should think more carefully what are their direct observations, what are their interpretations, etc, and adjust their presentations accordingly. This can help them to highlight their pure experimental data observations, which I believe is already interesting and striking enough and deserves to be published in a prestigious journal.

Reviewer #2:

Remarks to the Author:

Authors have revised the manuscript where the arguments are now much better supported by additional experimental and theoretical evidences. The magnetoelectric effects are clearly presented, and a demonstration of the multi-state data storage implemented by the ME effect is unique and hence guarantees the novelty of this work. Authors emphasized that the single layer SrOCl also host essentially the same phenomena, which could be also important. By the way, the explanation about the sample characterization is missing. How do authors confirm the single layer thickness? Furthermore, a comparison with thicker SrOCl will be also important. Are the magnetic and electric states maintained the same for the single layer SrOCl? And, would the explanation about the observed phenomena remain the same? I believe that the manuscript will be acceptable after adding the discussion about the single layer SrOCl.

Responses to Reviewer #1

Comment: *First of all, I would like to thank the authors for their careful and extensive answers to my questions (also questions from the other two reviewers). Overall, the manuscript presents an interesting magnetoelectric (ME) coupling in 2D CrOCl, where electric field can be used to control the magnetic order. Therefore, I support the publication of the paper.*

Response: We thank the reviewer for her/his careful reading and positive evaluations and are glad that the reviewer supports the publication of our paper. According to the reviewer's suggestions, we have revised our manuscript and provided the response in the following.

Comment: *Regarding some details of the reply, however, I would like to give some replies to the authors and perhaps also give some suggestions about how to improve the manuscript. I don't think the authors have presented direct evidence for the antiferroelectric (AFE) order. Only the two fold lattice distortion has been previously observed by neutron and XRD. But the two fold lattice distortion is only a prerequisite for the AFE order. The AFE order itself, remains to be directly confirmed. On the other hand, I don't think the AFE order is crucial for the key experimental observations, which is the ME coupling, i.e., electric field can directly control the magnetism in 2D CrOCl. I understand that, in the current understanding, the authors need the AFE order to explain the mechanism for the ME coupling: the authors believe that E-field couples to the AFE order, which in turn couples to the magnetism. However, to me, that is only an interpretation not a direct observation. Therefore, I suggest them to tone down their claim of the AFE order. They can focus on presenting their observations of how E and B fields control the magnetic order. Then they can suggest that the observed ME coupling is due to the existence of an AFE order. Overall, the authors should think more carefully what are their direct observations, what are their interpretations, etc, and adjust their presentations accordingly. This can help them to highlight their pure experimental data observations, which I believe is already interesting and striking enough and deserves to be published in a prestigious journal.*

Response: We sincerely appreciate the reviewer's suggestion, which is really helpful

for the rigor of our paper. As the reviewer has pointed out, the observation of the two-fold lattice distortion doesn't guarantee that the external field can generate a net polarization by tilting the antiferroelectric dipoles. Our central experimental observation is that the structural phase transition makes an additional contribution to the dielectric constant, which gives rise to tunable resistance states coupled to the magnetic phase transition. The phenomenological picture that the external electric field adjust the resistance states by tilting the antiferroelectric dipoles, while seemingly the most plausible explanation, remains to be directly confirmed by future high-resolution structural characterizations. Meanwhile, other analogous mechanisms such as the enhancement of the electron orbital polarization in the structural distorted phase may also lead to the additional net polarization and account for the magnetoelectric coupling effect. In the revised manuscript, we have weakened the demonstration of the tunability of AFE dipoles and discuss the possible explanations mainly based on purely experimental observations. We hope the revised manuscript will be satisfactory for publication.

Responses to Reviewer #2

Comments: Authors have revised the manuscript where the arguments are now much better supported by additional experimental and theoretical evidences. The magnetoelectric effects are clearly presented, and a demonstration of the multi-state data storage implemented by the ME effect is unique and hence guarantees the novelty of this work. Authors emphasized that the single layer SrOCl also host essentially the same phenomena, which could be also important. By the way, the explanation about the sample characterization is missing. How do authors confirm the single layer thickness? Furthermore, a comparison with thicker SrOCl will be also important. Are the magnetic and electric states maintained the same for the single layer SrOCl? And, would the explanation about the observed phenomena remain the same? I believe that the manuscript will be acceptable after adding the discussion about the single layer SrOCl.

Response: We thank the reviewer for her/his careful reading and positive evaluations. According to the reviewer's concerns about single-layer CrOCl, we provide the experimental data and discussions in the following.

Figure R1. Characterizations of single-layer CrOCl. (a) Angle-dependent tunneling currents under different external fields. The external field is rotated along an in-plane axis to the c -axis. θ is the angle between the external field and the in-plane axis. (b) Tunneling current

versus the external field along the *c*-axis. The transition fields of CrOCl are labelled by the black arrows, where H_1 and H_2 mark the beginning and the end of the spin-flop transition and H_3 marks the transition to the $\uparrow\uparrow\downarrow\downarrow$ phase. The inset shows the atomic force microscope height image of the exfoliated single-layer CrOCl. The black scale bar is 1 μm . (c) The tunneling current on the dependence of the in-plane field direction at different external fields. (d) Tunneling current *versus* in-plane field as the field points towards different in-plane directions. The transition field gradually increases as the external field deviates from the *a*-axis. All the data were obtained in the single-layer tunneling device (device 2 in the manuscript) with a bias voltage of -0.018 V .

We identified the exfoliated single-layer CrOCl by atomic force microscopy. An atomic force microscopy height image of a single-layer CrOCl flake which was used to construct the tunneling device (device 2 in the manuscript) is shown in the inset of Fig. R1b. The thickness of the flake is $\sim 0.7\text{ nm}$, in accordance with the DFT calculations^{1,2} of the layer distance and the previous report³. In order to confirm that the magnetic ground state of CrOCl persists down to the monolayer, we performed a series of tunneling current measurements similar to the multi-layer devices. The measured data of device 2 (the same single-layer device as in the manuscript) is presented in Fig. R1. As shown in Fig. R1a, when the external field was rotated from an in-plane axis to the *c*-axis, the tunneling current shows a 180° periodic symmetry. The magnetic phase transitions occur at the lowest field when the external field points to the *c*-axis, evidencing that the easy axis of single-layer CrOCl is still out-of-plane. The $I - B$ curve under the out-of-plane field is shown in Fig. R1b, where the spin-flop transition and the transition to the $\uparrow\uparrow\downarrow\downarrow$ phase can be clearly resolved. The transition fields are labelled as H_1 , H_2 and H_3 , which mark the beginning, the end of the spin-flop transition and the transition to the $\uparrow\uparrow\downarrow\downarrow$ phase, respectively. The values of the transition fields are also close to our previous results of the multi-layer tunneling devices⁴.

The response of the tunneling current to the in-plane field was also measured, presented in Fig. R1c and d. When the external field is rotated in the *ab*-plane of the sample, the tunneling current shows an exactly 180° periodic symmetry (Fig. R1c).

¹ Miao N, Xu B, Zhu L, et al. 2D intrinsic ferromagnets from van der Waals antiferromagnets[J]. Journal of the American Chemical Society, 2018, 140(7): 2417-2420.

² Zhang F, Kong Y C, Pang R, et al. Super-exchange theory for polyvalent anion magnets[J]. New Journal of Physics, 2019, 21(5): 053033.

³ Zhang T, Wang Y, Li H, et al. Magnetism and optical anisotropy in van der Waals antiferromagnetic insulator CrOCl[J]. ACS nano, 2019, 13(10): 11353-11362.

⁴ Gu P, Sun Y, Wang C, et al. Magnetic phase transitions and magnetoelastic coupling in a two-dimensional stripy antiferromagnet[J]. Nano Letters, 2022, 22(3): 1233-1241.

This evidences the persistence of the C_2 in-plane symmetry of single-layer CrOCl, characteristic of the one-dimensional stripy magnetic order. When the field points to the a -axis, the transition to the $a\uparrow\uparrow\downarrow\downarrow$ phase occurs at ~ 5.5 T and when the field direction rotates toward the b -axis, the transition field gradually increases and finally vanishes in our measurable field range. In summary, all the transition behaviors of the single-layer CrOCl are consistent with the reported multi-layer samples⁴. Our DFT calculations of the magnetic ground state and the resulted atomic distortions are also performed both in bulk and single-layer CrOCl and obtained the same results (shown in Fig. S1 in the Supplementary Information). Consequently, we conclude that the magnetic order, as well as the spin-induced electric transition, maintains the same for single-layer CrOCl.

Figure R2. Magnetolectric coupling in a single-layer CrOCl tunneling device (device 2). (a) $I - B$ curve of the CrOCl tunneling device at 2 K with the external field along with the a -axis. The pink curve shows the differential current $\delta I = I(B_{down}) - I(B_{up})$. The critical field where δI reaches a maximum is annotated by the black arrow. (b, c) Tunneling current (b) after alternating electric and magnetic excitations. The corresponding magnetic and electric fields *versus* time are plotted in (c). The device was parallel-connected with a 10 M Ω protection resistor.

The manipulation of the electric states in single-layer CrOCl is presented in Fig. R2. Similar to the multi-layer device reported in the manuscript, we obtained the maximum hysteresis at $B_0 = 5.72$ T by subtracting the current of the B -up curve from that of the B -down curve. Likewise, by alternately applying the magnetic and electric excitations, we can repeatedly realize the manipulation of the resistance state between two values, 225.5 k Ω and 227.5 k Ω . As a result, the same operation principles, as well as the microscopic mechanisms, apply to single-layer CrOCl. The only distinction is that in the single-layer device, the difference between the highest and lowest resistance states is lower, so it is difficult to realize multi-level resistance in a single device. This is probably caused by two reasons. Firstly, the tunneling

barrier of the single-layer CrOCl is much thinner, naturally resulting in lower magnetoresistance. Secondly, the electric dipoles in a monolayer can only host in-plane interactions with each other in the absence of neighboring layers, which may produce less metastable states during the phase transition. Nevertheless, the underlying physical picture remains the same. The characterizations of the single-layer CrOCl device, together with the discussion on the magnetoelectric coupling effect, have now been added to the revised Supplementary Information, and the manuscript has been revised correspondingly.

Reviewers' Comments:

Reviewer #1:

Remarks to the Author:

I wish to thank the authors for their answers. All my questions are addressed. I support publication.

Reviewer #2:

Remarks to the Author:

Authors replied my comments satisfactorily, and revised the manuscript accordingly. I now recommend the publication of this nice work in Nature Communications.

Responses to Reviewers

Reviewer #1: *“I wish to thank the authors for their answers. All my questions are addressed. I support publication.”*

Reviewer #2: *“Authors replied my comments satisfactorily, and revised the manuscript accordingly. I now recommend the publication of this nice work in Nature Communications.”*

Responses: We are glad that all reviewers are satisfied with the revision and agree to publish our work in *Nature Communications*. We would like to sincerely thank the reviewers, as our manuscript has been improved a lot thanks to the constructive comments from them. All their concerns are crucial for improving the quality and scientific rigor of our work.